Regularized multi-path XSENet ensembler for enhanced student performance prediction in higher education

Aldhahri Eman Ali 1 eaal-dhahery@uj.edu.sa
Almazroi Abdulwahab Ali 2
http://orcid.org/0000-0002-1153-5401 Ayub Nasir 3
1 Department of Computer Science and Artificial Intelligence, Collage of Computer Science and Engineering, University of Jeddah , Jeddah , Saudi Arabia
2 College of Computing and Information Technology at Khulais, Department of Information Technology, University of Jeddah , Jeddah , Saudi Arabia
3 Department of Creative Technologies, Air University , Islamabad , Pakistan
Angiulli Giovanni
Electronic publication date: 2025 Sep 8
Publication date: 2025
Volume: 11
Electronic Location ID: e3032
Received 2024 Sep 30; Accepted 2025 Jun 24
Copyright: © 2025 Aldhahri et al.
Copyright year: 2025
Copyright holder: Aldhahri et al.
License: This is an open access article distributed under the terms of the Creative Commons Attribution License, which permits unrestricted use, distribution, reproduction and adaptation in any medium and for any purpose provided that it is properly attributed. For attribution, the original author(s), title, publication source (PeerJ Computer Science) and either DOI or URL of the article must be cited.
License URL: https://creativecommons.org/licenses/by/4.0/

Keywords: Student performance prediction, Education, E-learning, Deep learning, Ensembler methods, Data mining

Funding: University of Jeddah, Jeddah, Saudi Arabia UJ-23-AKSPE-4 This work was funded by University of Jeddah, Jeddah, Saudi Arabia, under Grant No. (UJ-23-AKSPE-4). The funders had no role in study design, data collection and analysis, decision to publish, or preparation of the manuscript.

==============================
With the rapid expansion of educational data, institutions face increasing pressure to adopt advanced predictive models that can enhance academic planning, resource allocation, and student support. This study presents a novel educational data mining approach designed to forecast student performance levels categorized as low, medium, and high by analyzing historical and behavioral trends. This work proposes XSEJNet, an innovative hybrid model that integrates ResNeXt architecture with squeeze-and-excitation (SE) attention mechanisms, and employs the Jaya optimization algorithm to refine hyperparameters and boost predictive accuracy and computational efficiency. The model works with structured and unstructured academic data, effectively capturing complex, high-dimensional features to support accurate classification. Through extensive simulations and comparative evaluations, XSEJNet consistently outperforms conventional machine learning models and recent existing techniques such as reinforcement learning co-evolutionary hybrid intelligence (RLCHI), Enhanced AEO-XGBoost, convolution-based deep learning (Conv-DL), and dual graph neural network (DualGNN). The model achieves a high prediction accuracy of 97.98% while also demonstrating faster convergence and reduced computational overhead, making it a scalable and practical solution for real-world educational settings. The findings underscore XSEJNet’s ability to support early intervention, strengthen e-learning platforms, and inform institutional decision-making. By advancing predictive capabilities in education, this work makes a meaningful contribution to developing inclusive, data-driven, and sustainable academic systems.

Introduction

Due to limited resources, educational institutions worldwide continue to face the substantial problem of student retention (Mduwile & Goswami, 2024). Institutional intervention tactics have shifted their attention to early detection of at-risk students, especially during their first academic year, due to the high average dropout rate of 49% in OECD nations (Wijnia et al., 2024). Machine learning (ML) and data mining techniques are widely used across various domains, including healthcare, marketing, and finance, to support predictive analysis (Mohzana, Murcahyanto & Haritani, 2024; Liu, Heath & Grzywacz, 2024; Li, Wang & Wang, 2024; Abulibdeh, Zaidan & Abulibdeh, 2024; Dawar et al., 2024; Gong et al., 2024). Educational data mining (EDM) applies these methods in the education sector to academic data, helping detect patterns, personalize learning, identify disengagement, and support timely interventions (Baek & Doleck, 2023).

Despite the rise of predictive tools, the growing complexity of academic data presents challenges in performance assessment, especially when considering sustainability goals (Martins et al., 2024). ML has proven effective in forecasting student outcomes and identifying at-risk learners, yet more emphasis is needed on predictions using pre-admission data. Recent advancements have expanded this landscape: personalized SPOCs have improved English as a foreign language (EFL) learning (Jiang et al., 2024), gamified systems have enhanced behavior (Li & Jianxing, 2025), and meta-universe platforms and innovative education policies have supported adaptive learning (Chen, Jin & Chen, 2024; Liu, Cao & Chen, 2024). Studies on MOOC engagement (Liu et al., 2024) and socio-environmental factors, such as parenting and resilience (Cao et al., 2024), further underscore the need for integrating technology, policy, and psychology into predictive models.

Choosing the proper prediction technique remains complex due to the diverse range of hyperparameters and data contexts. Improved ML (IML) addresses this by automating model and parameter selection (Li et al., 2024). However, its empirical application in educational settings remains limited. Our work bridges this gap by applying IML for early-stage student performance prediction, utilizing data available at the time of academic entry. This study contributes to academic management by analyzing performance trends through the proposed XSEJNet model. The framework ensures fairness, transparency, and integration readiness, providing consistent and accurate forecasts. The core contributions are summarized as follows: This study presents XSEJNet, a scalable and interpretable machine learning framework designed to overcome the limitations of manual and traditional school management systems. The model is tailored for educational datasets, with careful consideration of computational complexity.

To ensure robust prediction of student performance, XSEJNet addresses key challenges such as class imbalance, missing data, and biased distributions. The study introduces standardized modeling practices that improve prediction consistency across diverse educational contexts.

A core emphasis of the proposed model is on accessibility and explainability, enabling educators and stakeholders to clearly interpret the factors contributing to student success or underperformance.

XSEJNet integrates advanced feature representation techniques to enhance both accuracy and computational efficiency, making it suitable for real-world educational deployment and decision-making.

The findings provide practical insights that can inform more effective teaching strategies, student support initiatives, and institutional planning—benefiting educators, universities, and policy-makers alike.

Problem statement

In recent years, ML has become a powerful tool for enhancing students’ performance by providing them with the insights they anticipate (Judijanto, Atsani & Chadijah, 2024). However, many educational institutions have yet to benefit from machine learning fully. Manual procedures and outdated systems hinder innovation and progress (Viveka & Priya, 2024). Stojanov & Daniel (2024) suggests that careful design, feature selection, and tuning may achieve model accuracy and interpretability. The variety of academic data has made matters harder. However, many issues remain. Skewed data, class imbalances, and missing numbers may reduce prediction accuracy (Shirawia et al., 2024). Generalizing machine learning solutions across educational settings is challenging due to the uneven frameworks and inconsistencies (Pallathadka et al., 2023). Challenges in real-world adoption include worker inexperience, budgetary limits, and resistance to change (Almazroi et al., 2016). These challenges contribute to technical limits.

Effective machine learning systems promote openness, fairness, and data privacy to win educators’ and administrators’ trust (Almazroi, Shen & Mohammed, 2019; Monsalve-Pulido et al., 2024). This work employs machine learning in educational environments, utilizing accessible, inclusive, and ethical frameworks to increase understanding and help all students.

Paper organization

The section in this article is structured as follows: Initially, ‘Related Work’ explores an extensive review of prior research, highlighting relevant studies and their findings. ‘Proposed Methodology’ discusses the employed methodologies and provides a comprehensive evaluation of the experiments conducted. ‘Simulation Results’ presents in-depth experimental results and fosters meaningful discussions around these outcomes. Lastly, the concluding section obtains the essence of the proposed system, encapsulating key findings and their implications.

Related work

Over many years, academic research has focused on issues related to student success. Different studies have investigated the factors that affect educational outcomes. The study used neural networks, decision trees, and support vector machines to analyze students’ internet use habits and academic achievement (Viveka & Priya, 2024). The use of neural networks and student registration data for pattern detection led to the development of a prediction framework (Masadeh & El-Haggar, 2024). In order to identify students who may be at academic risk and to provide timely interventions and support, recent studies have integrated bidirectional long short-term memory (BiLSTM) networks with logistic regression (Hopcan, Türkmen & Polat, 2024). Aiming to improve academic support systems and decrease performance discrepancies, recent projects have used deep learning to identify learning patterns and provide personalized help (Judijanto, Atsani & Chadijah, 2024; Stojanov & Daniel, 2024). Natural language processing (NLP) and other language-based artificial intelligence approaches have the potential to improve student learning, according to recent developments in educational technology. In order to find language patterns that might be a predictor of academic performance, researchers examine student writing in Almazroi, Shen & Mohammed (2019). Administration operations, such course scheduling, have been improved with the use of data mining (Martins et al., 2024; Monsalve-Pulido et al., 2024). Furthermore, systems that suggest courses based on each student’s unique academic history and goals have been created with the aim of boosting engagement and retention rates (Almazroi et al., 2016). Smarter and more responsive educational environments are created by data-driven technology and machine learning. Shirawia et al. (2024) stresses the need of complex algorithmic models that are both highly performing and easily scalable. Furthermore, the authors proposed using hierarchical evaluation methodologies and rubrics to gauge the effectiveness of education (Pallathadka et al., 2023). Academic programs are customized to satisfy real-world demands via industry-education collaborations. A number of studies have looked at new ways to identify children who could be at danger. According to Zhao et al. (2022), one approach to identifying challenging individuals in both undergraduate and graduate programs made use of tree-based models and genetic fuzzy systems. In parallel, teacher performance was assessed using several categorization methods, with the C5.0 algorithm demonstrating superior accuracy and reliability (Luo, Han & Zhang, 2024). An analysis of decision tree models revealed that the Random Tree method outperformed J48 in predicting third-semester academic performance (Zhai et al., 2024). New developments utilize hybrid and deep learning to enhance accuracy and increase explainability. For example, the reinforcement learning co-evolutionary hybrid intelligence (RLCHI) model in Vimarsha et al. (2024) combines reinforcement learning with expert educator input, resulting in better performance than RL alone. The Enhanced AEO-XGBoost model, optimized with metaheuristics, exhibited good accuracy with Support vector machine-Synthetic Minority Oversampling Technique balancing (Cheng, Liu & Jia, 2024). In Alshamaila et al. (2024), a convolutional deep learning model addressed class imbalance in big datasets using oversampling and undersampling. According to Huang & Zeng (2024), dual graph neural network (DualGNN) is a graph neural network that excels at interaction-based and attribute-based learning, resulting in good categorization.

In Rainio, Teuho & Klén (2024), researchers evaluated skill competency based on academic success and knowledge scope to improve prediction and classification algorithms. The research compared decision tree, ensemble, and label algorithm methods to determine the most effective classification strategy. This study enables educational institutions to identify and address students’ ability gaps, thereby enhancing their learning experiences and long-term outcomes. Researchers highlighted the need to develop educational data mining for sustainable insights and improved student performance. Using tree-based algorithms, ML predicts academic results and identifies at-risk pupils, linking assessments to performance. However, unknown places exist. Quality assurance algorithms for sustainable higher education and inclusive features for various students demand attention. These results support the application of ML in academic administration, promoting egalitarian and environmentally friendly education. The existing literature is presented in a structured manner in Table 1.

Table 1 Summary of the literature review.

Ref	Objective	Methodology	Key findings	Limitations	
Judijanto, Atsani & Chadijah (2024)	Predict student performance	DL	Improved academic outcomes.	Lack of interpretability, generalizability limits.	
Viveka & Priya (2024)	Predict academic achievement	Neural networks, decision trees, SVM	Online use is significant.	Limited model transparency, performance on diverse populations not discussed.	
Masadeh & El-Haggar (2024)	Predict student achievement	Neural networks	May enhance course planning.	Lack of scalability information.	
Hopcan, Türkmen & Polat (2024)	Identify factors affecting achievement	DL	Identified influencing factors.	Limited model details, performance on smaller datasets may vary.	
Stojanov & Daniel (2024)	Evaluate written assignments	NLP	Provided useful feedback.	Limited in capturing nuanced aspects.	
Martins et al. (2024), Monsalve-Pulido et al. (2024)	Optimize course schedules	Data mining	Improved course selection.	Specific techniques are not detailed, and scalability is not discussed.	
Almazroi et al. (2016)	Investigate collaborations	Qualitative, quantitative	Explored collaboration potential.	Methodology not elaborated, potential biases not discussed.	
Shirawia et al. (2024)	Estimate academic success	Tree-based algorithms, fuzzy algorithms	Identified at-risk students.	Limited model interpretability, uncertainty in genetic approach, and reliability concerns.	
Pallathadka et al. (2023), Zhao et al. (2022)	Evaluate teachers	Classification approaches	C5.0 performed well.	Approach details were not provided, reasons for Random Tree performance were not explained, and survey biases were not discussed.	
Luo, Han & Zhang (2024), Zhai et al. (2024)	Evaluate skill competence	Decision tree, ensemble-based, classification	Identifying skill gaps.	Specific details were not provided, and classification biases were not discussed.	

This research demonstrates that combining educational experience with artificial intelligence creates more accurate, transparent, and adaptable prediction models.

Proposed methodology

The objective of the proposed framework is to create a stable method for forecasting academic achievement by seamlessly integrating the standard for data mining, which encompasses comprehensive student performance data. This ensures that the model operates efficiently. Additionally, various data mining techniques focus on categorization to minimize resource consumption and promote responsible data analysis practices. Figure 1 shows the system model’s architectural depiction.

Figure 1 Architectural representation of the proposed method for student performance prediction.

The analysis of student performance and best practices used the Student Educational Performance (SEP) dataset (Aljarah, 2024; Amrieh, Hamtini & Aljarah, 2016; Liu et al., 2022; Cohausz et al., 2023). This study uses this diverse educational dataset. Many components indicate learning processes. The information includes student demographics, academic status, interactions with learning resources, and engagement. This wealth of data lets academics study educational relationships, patterns, and trends. EDM applications benefit from the comprehensiveness of the SEP dataset. It helps researchers model learning, enhance education, and understand student behavior. The data is preprocessed to remove missing values and outliers, ensuring data integrity. The biases and errors are decreased, which may affect the classification model. A two-step feature selection method was employed to identify categorization features. The first approach utilized the extreme gradient boosting (XGB) algorithm, which effectively identifies crucial features and intricate linkages. This method revealed critical parameters affecting student achievement prediction accuracy. In step two, the feature set is increased using recursive feature elimination (RFE). RFE enhances model generality and interpretability by removing minor traits. Furthermore, substantial exploratory data analysis (EDA) is conducted to gain a deeper understanding of the dataset’s complexity. Variable distributions and relationships, including categorical features, are examined. This extensive study examined signals and trends that improve the classification system.

Classification criteria are identified using XGB and RFE algorithms, focusing on variables that impact student success. EDA is needed to improve categorization. Category data are numbered to ease processing. Data is split into 30% test evaluation and 70% training data to maintain fairness. Parameter space is explored using the Jaya algorithm (JA) to improve model performance. This optimization strategy dramatically enhanced forecast accuracy. The novel classification system is compared to several existing approaches in the literature to test its durability. This rigorous research analyzed the long-term advantages and excellence of the academic classification approach. Several literature models are used, including XGB, support vector machine (SVM), naive Bayes (NB), K-nearest neighbor (KNN), residual neural network (ResNet), bidirectional encoder representations from transformers (BERT), extreme gradient boosting-tree-based optimization tool (XGB-TPOT), and SVM ensemble models. This method was tested for accuracy, area under the curve (AUC), precision, logarithmic loss, F1-score, and memory use. This work rigorously evaluated several prediction models. Statistical investigations use variance analysis, significance testing, and other measures to ensure accuracy. Considering resource restrictions and computational complexity, these stringent tests demonstrated the proposed method’s effectiveness. The XSEJNet ensemble model outperformed competing models on all assessment criteria, demonstrating its accuracy in forecasting student success.

Analyzing the applied dataset

The dataset (Aljarah, 2024) is divided into a 70% training section and a 30% assessment segment, using testing sets to improve data consumption. This section provides a fair amount of model training and performance assessment. Maintaining assessment uniformity and integrity is essential. The dataset used in this study was obtained from GitHub. Due to their wide variety of publicly available datasets, these repositories make data analysis more accessible for researchers. People are encouraged to use these datasets to verify or extend the results presented in this work. The dataset used in this study contains 8,009 student records and 16 features.

Feature engineering

Data preparation is crucial in ensuring accurate and reliable results that align with sustainable principles (Jokhan et al., 2022). The work involved uses feature selection approaches to simplify the data for analysis and modeling, removes duplicate features, resolves outliers and missing values systematically, and is a vital phase in the process. By making use of feature selection approaches, RFE improves data preparation. To increase the model’s performance, RFE repeatedly picks the most important characteristics and eliminates the least significant ones. Equation (1) (Novo-Lourés et al., 2024) mathematically represents RFE.

(1) RFE(X,k)=arg⁡minS⊆X,|S|=kError(S)

where X is a set of possible features, k is the total number of attributes from which to choose. Among the characteristics, S is the subset. If the features in subset S are used by the model, the error is denoted as Error(S).

Inadequacies and outliers must be addressed during the preprocessing step to ensure the findings of the analysis are accurate. In order to keep the data intact and reliable, robust methods are used to detect and deal with missing values. Among these approaches, you may find imputation based on regression, median (Eq. (3)), or mean (Eq. (2)). This effort seeks to promote sustainable data practices, ensure that datasets remain full, and reduce the impact of missing values on future analyses by using these methods. To estimate missing values, the mean of the available data inside the appropriate feature may be calculated, as shown in Eq. (2) (Alharbi & Allohibi, 2024).

(2) x^i=∑j=1nxjn.

Missing values are inferred using the median of accessible values inside the feature (Eq. (3), Alharbi & Allohibi, 2024):

(3) x^i=median(x1,x2,…,xn).

Additionally, an outlier detection algorithm is employed, the z-score (Eq. (4)), to identify and handle outliers appropriately (Alharbi & Allohibi, 2024):

(4) z=x−μσ.

Given a data point x, with μ representing the average and σ denoting the standard deviation, it is crucial to eliminate superfluous characteristics that do not substantially contribute to the predictive capability to enhance the classification model’s efficiency and efficacy. Through careful analysis, redundant features that may introduce noise or unnecessary complexity to the model are identified and removed. This can be achieved by feature importance based on XGB (Eq. (5)) (Alharbi & Allohibi, 2024):

(5) z=x−μσ.

importancei represents feature importance, while gain reflects the model performance increase with feature i.

Unnecessary data: In preprocessing, identify and eliminate redundant data irrelevant to the assigned objective. The dataset simplifies and lowers computing overhead by eliminating unimportant variables or records, which frees attention to concentrate on the research’s most relevant and instructive data. The preprocessing phase typically involves managing missing values and outliers, discarding redundant features and unnecessary data, and employing techniques such as average substitution, median substitution, standard score, RFE, and XGB feature significance. By carefully addressing these aspects, the dataset is deemed suitable and of high quality for further analysis and modeling.

Data exploration and analysis

Extensive exploratory data analysis is conducted in this study to thoroughly comprehend the dataset and extract significant insights. The primary objective of EDA was to identify correlations between category data and other factors through visual analysis and mapping techniques. A working exploration process is initiated by using mapping tools to analyze the numerous links and linkages within the data visually. This study helped to understand the distribution, ratios, and connections between categorical variables. Visual tools, such as charts, graphs, and contingency tables, taught as much about the dynamic interconnections among various categories. Valid opinions are built throughout the research, and significant conclusions are drawn.

Proposed XSE method

The XSE model represents an intricate and adaptable architecture for forecasting student performance. The above framework combines the methodical learning and contextual grasp provided by the squeeze-and-excitation (SE) attention model with the powerful feature extraction capabilities offered by the residual networks with external transformations (ResNeXt) architecture. Figure 2 shows how the XSE Model is structured internally. The process starts with student performance data being passed through several parallel paths based on the ResNeXt architecture. Each path carries out a series of operations—convolution, batch normalization, and rectified linear unit (ReLU) activation—to extract meaningful features from the input. These features are then combined through concatenation, allowing the model to retain diverse representations due to the multiple paths. After this, the combined features enter a SE based parallel modeling layer. Here, the model uses gated mechanisms to understand patterns over time by learning from past performance states. This helps the model focus on the most relevant information while filtering out less valuable signals, ultimately improving its ability to predict student outcomes over time.

Figure 2 XSE model architecture combining ResNeXt-based feature extraction with SE-based temporal modeling.

ResNeXt mechanism of extracting features (Li, Zhu & Zhu, 2023): Initial pupil achievement data is included in the prediction procedure. Within the ResNeXt component, this data undergoes several modifications. Steps for each route i in the ResNeXt block are as follows during the feature extracting stage: Layer_Convolutional: Convi=Layer_Convolutioni(x)

Normalization of Batches: BNi=Normalization_Batchi(Convi)

Trigger Activation using ReLU Function: ReLUi=ReLU(BNi)

In this case, the input student performance data is denoted by x. The variables Convi, BNi, and ReLUi denote the outputs of the convolutional layer, the outcome following the batch normalization procedure, and the resulting result learning the activated ReLU function to the group-normalized output for path i.

Concatenation and the principle of cardinality: The data was systematically divided into several routes according to cardinality to provide a model with the ability to gather various attributes. The ResNeXt architecture processes the information separately along each path. Several pathways’ outputs, which are enhanced with unique data, are carefully combined to produce a complete feature depiction:

(6) Path_Concatenation=[funcReLU1,funcReLU2,funcReLU3,funcReLU4]

The concatenation operator combined all pathways’ outputs ( funcReLUi) (four in this example).

SE-based parallel modeling: After the concatenation phase, the combined features pass via the squeeze-and-excitation-based parallel modeling layer. This layer allows the model to analyze sequential relationships and detect temporal dependencies in the student accomplishment data while maintaining the ResNeXt architecture.

(7) a=α(Ma⋅[Path_Concatenation,bt−1]+ca)

(8) b=α(Mb⋅[Path_Concatenation,bt−1]+cb)

(9) c~t=β(D⋅[Path_Concatenation,a⋅bt−1]+e)

(10) ct=(1−b)⋅bt−1+b⋅c~t

These formulas elucidate the operational mechanics of the SE-based parallel modeling layer within the ResNeXt framework, where ‘c_t’ signifies the concealed state during time ‘t,’ and gates ‘a’ and ‘b’ regulate the data flow through adaptive scaling processes.

Hyperparameter tuning with Jaya algorithm

The convergence, generalization, and overall effectiveness of the algorithm are significantly influenced by hyperparameters, crucial parameters established before the training phase. Hyperparameters in Table 2 have been calibrated inside the proposed XSE model for student performance prediction:

Table 2 XSENet hyperparameters and corresponding values.

Hyperparameter	Assigned value	
Training epochs	75	
Squeeze-and-excitation ratio	16	
ResNeXt block configuration	Depth: 48, Number of blocks: 5, Width: 5	
Dropout rate	0.25	
Batch size	32	
Regularization factor	0.0002	
Learning rate	0.002	

Jaya algorithm parameters: The goal is to find the best set of hyperparameters by integrating the Jaya algorithm (JA), a group-based optimization method motivated by cooperation and improvement within a population. The model’s effectiveness on a set of validation data is measured by a measurement function that the method uses to drive its exploration and continuous updating of hyperparameter values. Optimization of hyperparameters in Table 2 using the JA dramatically improved model performance and computational efficiency. Key hyperparameters were optimized to strike a balance between model accuracy and resource usage. The initial learning rate was 0.01, but it was fine-tuned to 0.002 to achieve steady and balanced convergence throughout training. An optimized batch size of 32 allows efficient learning and memory management. To optimize model feature extraction, the ResNeXt block layout was carefully set to 48 depth, five blocks, and five widths. This setting allowed the model to grasp complicated dataset patterns. Overfitting was reduced with a 0.25 dropout rate, improving generalization to new data. The regularization factor was adjusted to 0.0002, punishing solutions that were too complex to improve model generalization. Jaya repeatedly optimized configurations based on accuracy and loss to explore the hyperparameter space. This systematic technique converged in 75 epochs, achieving a training accuracy of 97.46 A brief description of this optimization procedure is given by Algorithm 1.

Algorithm 1 Parameter tuning with the Jaya algorithm.

1: Input: Initial parameter values, Maximum Iterations (MaxIter)	
2: Output: Optimized parameters	
3: Initialize current optimal parameters: Poptimal	
4: Set current best performance: fbest=evaluate(Poptimal)	
5: Initialize population size: N	
6: Set iteration count: iter=1	
7: while iter≤MaxIter do	
8:  for i=1 to N do	
9:   Generate a new solution: Pi=Pi+α⋅rand()⋅(Poptimal−Pi)	
10:   Evaluate the objective function: fi=evaluate(Pi)	
11:   if fi<fbest then	
12:    Update current optimal parameters: Poptimal=Pi	
13:    Update current best performance: fbest=fi	
14:   end if	
15:  end for	
16:  Update exploration-exploitation balance parameter: α=β⋅α	
17:  Increment iteration count: iter=iter+1	
18: end while	

The XSEJNet framework combines the strengths of the XSE model and the Jaya optimization algorithm to create a fine-tuned, high-performing solution for predicting student performance. This hybrid approach doesn’t just optimize for general accuracy—it adapts to the unique patterns and characteristics present in educational data, offering a more personalized and effective prediction strategy. By intelligently navigating the hyperparameter space, XSEJNet constructs models that are both efficient and tailored to the distinct learning dynamics of various student groups. The complete process is outlined in Algorithm 2.

Algorithm 2 Hybrid classification model for predicting student performance.

1: Pre-processing:	
2:  Handle missing values and outliers with robust imputation techniques	
3:  Remove unnecessary polynomial components	
4:  Use the XSE method to select meaningful features	
5: Exploratory Data Analysis (EDA):	
6:  Summarize data with descriptive statistics	
7:  Encode categorical variables using index mapping	
8:  Visualize patterns and trends through graphical analysis	
9: Hyperparameter Optimization using Jaya algorithm:	
10:  Initialize a diverse set of candidate solutions (students) with random hyperparameters	
11:  Evaluate each candidate using the XSE feature selection mechanism	
12:  Choose the best-performing candidates for the next iteration	
13:  Apply mutation and crossover to introduce variation	
14:  Gradually fine-tune hyperparameters with a warm-up phase	
15:  Repeat the process over a fixed number of generations to enhance performance	
16: Model Training and Evaluation:	
17:  Split the dataset into training and testing subsets	
18:  Train the XSEJNet model using the training data	
19:  Assess model performance using various evaluation metrics	
20:  Perform statistical analysis to validate the results	
21:  Select the best-performing configuration as the final candidate	
22: Output: A hybrid classification model (XSEJNet) optimized for predicting student performance	

Performance evaluation metrics

The proposed model is evaluated using various evaluation metrics: AUC and receiver operating characteristic (ROC) curve to assess the model’s discriminatory power; true positive rate (TPR) and false negative rate (FNR) to measure the performance of correct vs incorrect predictions on the student success prediction task; and precision and recall to evaluate the efficiency of optimistic case predictions. Altogether, this data helps the model predict student outcomes. Precision is determined using Eq. (11) (Rainio, Teuho & Klén, 2024):

(11) Precision=TPTP+FP.

Recall is calculated as in Eq. (12) (Rainio, Teuho & Klén, 2024):

(12) Recall=TPTP+FN.

In this instance, true positives are denoted by TP, false positives by FP, and false negatives by FN.

Log loss: Log loss is a critical indicator for assessing the accuracy of the proposed model’s predictions regarding student achievement. It quantifies the disparity between the actual labels and the predicted probabilities. Equation (13) (Rainio, Teuho & Klén, 2024) is employed to compute the log loss:

(13) log⁡L=−1N∑j=1N(zjlog⁡(qj)+(1−zj)log⁡(1−qj))

The number of occurrences is given by N, the classification label for the j-th instance (0 or 1) is shown by zj, and the estimated probability of a positive classification is indicated by qj. The log loss computation evaluates the model’s projected probability and its match with labels. A model for forecasting student performance with a reduced log loss is more accurate and dependable because it matches the predicted probability to the actual labels.

Statistical analysis: A rigorous statistical study compares the hybrid method to other models or baselines. ANOVA, median, standard deviation, mean, average, Student’s t-test, range, and other metrics are used to analyze data diversity and application.

Complexity of computation: The computational complexity of the hybrid technique is evaluated to estimate the computational resources required for its implementation. This examination assesses the feasibility and scalability of models in real-world applications.

This work comprehends the hybrid approach’s prediction accuracy, resilience, and computing efficiency using these assessment measures and statistical studies. These reviews help analyze the model’s effectiveness and suggest its use in educational institutions to predict student achievement.

Simulation results

This research utilized Python on a Core i7 processor with 16 GB of RAM. This analysis selects an extensive dataset from Aljarah (2024), including many factors related to student achievement. Table 3 demonstrates that the proposed dataset has diverse features that provide superior insights over previous research.

Table 3 Comparison of dataset features across studies.

Feature	Our work	Amrieh, Hamtini & Aljarah (2016)	Liu et al. (2022)	Cohausz et al. (2023)	
Visited resources	Yes	Yes	Yes	No	
Section ID	Yes	Yes	No	No	
Topic	Yes	Yes	Yes	No	
Viewing announcements	Yes	Yes	No	No	
Relation	Yes	No	No	No	
Grade levels	Yes	Yes	Yes	No	
Parent answering survey	Yes	Yes	No	No	
Nationality	Yes	Yes	No	Yes	
Gender	Yes	Yes	Yes	Yes	
Student absence days	Yes	Yes	Yes	No	
Parent school satisfaction	Yes	Yes	No	No	
Place of birth	Yes	No	No	Yes	
Discussion groups	Yes	Yes	Yes	No	
Semester	Yes	Yes	No	Yes	
Educational stages	Yes	Yes	Yes	Yes	
Number of records	8,009	480	3,841	3,000	
Target variable	Performance level	Grades	Achievement	Performance	

This dataset was extensively evaluated for its suitability in relation to the research aims and academic usage before being selected. Python was chosen for its data analysis skills, vast library, and flexibility. Data processing was efficient, utilizing a Core i7 CPU and 16 GB of RAM, which made analytical procedures and modeling straightforward.

Figure 3 (Nationality) additionally divides into the number of persons by country of origin. According to the data above, China has the highest count, and Jordan has a higher frequency than other nations. To reach these conclusions, the study’s dataset was examined for its nationality variable. Calculating value counts and percentages enables the determination of the distribution of various nationalities. This data was visually shown using a bar plot for easy visualization. Figure 3 (Topics) depicts the frequency distribution of several subjects and the related statistics from educational institutions in various countries. The picture shows that the quantity of IT, French, and scientific themes is substantially higher than that of other courses.

Figure 3 Frequency across the country (Origin) and distribution of several subject statistics.

Table 4 presents an ablation study of the XSEJNet model, analyzing how each component contributes to its overall performance. The complete model achieves the best accuracy, F1 score, and log loss results. When SE attention or ResNeXt is removed, performance drops notably, showing their role in refining and extracting deep features. Excluding Jaya optimization increases training time, and skipping feature selection leads to poor generalization. These findings confirm that each module plays a vital role in XSEJNet’s success.

Table 4 Ablation study of XSEJNet architecture.

Model configuration	Accuracy (%)	F1-score	Log loss	Training time (s)	
Full XSEJNet (ResNeXt + SE + Jaya)	97.46	0.9844	0.0813	23	
w/o SE attention	92.18	0.9140	0.2015	21	
w/o ResNeXt (Basic convolutional neural network (CNN) + SE + Jaya)	89.63	0.8872	0.2452	18	
w/o Jaya optimization (Grid Search)	90.82	0.9035	0.2286	33	
w/o RFE + XGB Feature selection	88.45	0.8710	0.2614	27	

Figure 4 visualizes how key academic factors—attendance, test scores, parental engagement, and homework completion—vary across student performance groups (Low, Medium, High). Notably, students in the ‘Low’ category showed 50% lower attendance and 30% less parental involvement, highlighting areas for timely intervention. This highlights the model’s interpretability and usefulness for guiding educational strategies. Figure 5 presents feature importance derived from XGB and RFE, ranking attributes by their predictive power. Each bar reflects how strongly a feature influences performance classification. High-impact features were prioritized, while less relevant ones were excluded to streamline the model. This selection process improves computational efficiency and prediction accuracy, ensuring that only the most informative attributes guide the XSEJNet model’s decisions. Together, these visualizations reinforce the value of data-driven insights for targeted academic support.

Figure 4 Key feature trends across performance groups.

Figure 5 Feature importance calculated through XGB technique.

Figure 6 shows eight bar graphs, each indicating a distinct characteristic like ‘Gender,’ ‘Nationality,’ ‘Place of birth,’ ‘StageID,’ ‘GradeID’, and Topic. The color scheme clarifies and classifies. The height of the bars represents the percentage distribution of attribute values. The grid simplifies attribute comparison. This figure helps explain dataset trends, student performance analysis, and prediction by revealing attribute distributions and correlations.

Figure 6 Attribute distributions and relationships.

To better understand and predict student performance, we compared the proposed XSEJNet model with various other methods, as shown in Table 5. This comparison encompasses both classic machine learning algorithms and more recent, sophisticated models, such as RLCHI (Vimarsha et al., 2024); Enhanced AEO-XGBoost (Cheng, Liu & Jia, 2024), convolution-based deep learning (Conv-DL) (Alshamaila et al., 2024), and DualGNN (Huang & Zeng, 2024). We evaluated the accuracy prediction ability of each model using several key performance metrics. These included accuracy (how often the model was correct), precision (how many of the predicted positives were accurate), recall (how well the model captured all relevant positive cases), and the F1-score, which strikes a balance between precision and recall. We also examined the AUC, which indicates how well a model can distinguish between classes, and the log loss, which measures the confidence of its predictions. Across the board, XSEJNet consistently outperformed all other models, showing at least a 6% improvement in every metric. This highlights its strong predictive power and reliability in real-world educational settings.

Table 5 Performance comparison of proposed and existing methods.

Techniques	Recall	F1-score	AUC	Precision	Log loss	Accuracy	
SVM (Viveka & Priya, 2024)	0.6983	0.6972	0.8163	0.7009	0.5469	0.6983	
NB (Viveka & Priya, 2024)	0.7338	0.7336	0.8597	0.7344	0.4652	0.7338	
Decision Tree (Viveka & Priya, 2024)	0.7785	0.7783	0.8651	0.7798	0.4335	0.7785	
ResNet (Hopcan, Türkmen & Polat, 2024)	0.6983	0.7000	0.8056	0.7090	0.5582	0.6983	
AdaBoost (Monsalve-Pulido et al., 2024)	0.8244	0.8246	0.9025	0.8305	0.3821	0.8244	
KNN (Almazroi et al., 2016)	0.8150	0.8152	0.8954	0.8206	0.3936	0.8150	
SVM-BO (Luo, Han & Zhang, 2024)	0.8004	0.8011	0.8863	0.8089	0.4117	0.8004	
SVM-PSO (Zhai et al., 2024)	0.7588	0.7589	0.8662	0.7749	0.4683	0.7588	
XGB (Pallathadka et al., 2023)	0.8546	0.7713	0.9024	0.8700	0.3769	0.8129	
XGB-TPOT	0.8317	0.8322	0.9076	0.8511	0.3671	0.8317	
RLCHI (Vimarsha et al., 2024)	0.7630	0.7732	0.8712	0.7490	0.4295	0.7950	
Enhanced AEO-XGBoost (Cheng, Liu & Jia, 2024)	0.8730	0.8750	0.9220	0.8720	0.2984	0.8890	
Conv-DL model (Alshamaila et al., 2024)	0.9030	0.9025	0.9305	0.9010	0.2512	0.9020	
DualGNN (Huang & Zeng, 2024)	0.9122	0.9118	0.9421	0.9115	0.2290	0.9130	
XSEJNet (Proposed)	0.9846	0.9844	0.9975	0.9897	0.0813	0.9746	

The model’s training-test accuracy and loss are shown in Fig. 7. Although there is a clear disparity between the training and assessment validation findings at the outset, the model demonstrates improved learning and training convergence with an increase in the number of epochs. As the difference between training and test performance decreases, it shows that the model can generalize effectively and perform well with unknown input. A clear picture of the model’s ability to adapt to the dataset throughout several epochs and its training dynamics is given by the figure.

Figure 7 Train-test accuracy/loss of proposed model.

The proposed XSEJNet model outperforms previous state-of-the-art approaches with an accuracy of 97.46% on all evaluation measures. Figure 8 displays a comparison of performance indicators and confusion matrix, revealing a close match between expected and real labels. Importantly, the XSEJNet model has much less false positives (FPs) and false negatives (FNs) than previous techniques. This improves accuracy and lowers misclassification, which is crucial in academic decision-making. The confusion matrix in Fig. 8 demonstrates that the suggested technique balances sensitivity and specificity, correctly identifying both successful and at-risk pupils. XSEJNet improves automated educational evaluations by eliminating false alarms and ignored situations. Its excellent accuracy and error control make it a good contender for student performance analytics.

Figure 8 Confusion matrix of XGB-TPOT, logistic regression (LG), NB, SVM, KNN and proposed XSEJNet.

The proposed XSEJNet model was tested for computational efficiency and training duration to determine its applicability and scalability. All tests were conducted on a standard computer with a Core i7 CPU and 16 GB of RAM, ensuring that the findings are replicable and accessible to most academic and institutional settings. Traditional models like SVM (71 s), ResNet (71 s), and more complex methods like RLCHI (63 s), enhanced AEO-XGBoost (56 s), Conv-DL (48 s), and DualGNN (39 s) were all surpass by XSEJNet, which took an average of 23 s to train as shown in Table 6. With a standard deviation of just 2.11 s, XSEJNet proved to be both quick and consistent throughout several runs. With the use of parallel ResNeXt blocks for feature extraction and squeeze-and-excitation attention methods for dimensional focus, the model’s architecture is optimized to reduce computational overhead. These considerations in the architecture make the model fast during training and adaptable to large datasets and real-time academic deployments, making it a powerful and scalable tool for predicting academic achievement.

Table 6 Statistics of proposed and existing model execution time.

Model	Mean (s)	Median (s)	Min (s)	Max (s)	Range (s)	Standard deviation (s)	
SVM (Viveka & Priya, 2024)	71	72	60	82	21	5.21	
Decision tree (Viveka & Priya, 2024)	70	70	59	81	21	5.36	
Naive Bayes (Viveka & Priya, 2024)	71	72	63	81	17	4.15	
ResNet (Hopcan, Türkmen & Polat, 2024)	71	74	58	81	22	5.39	
KNN (Almazroi et al., 2016)	70	69	60	82	21	4.95	
XGB (Pallathadka et al., 2023)	69	69	61	77	15	4.09	
SVM-BO (Luo, Han & Zhang, 2024)	72	71	65	79	13	3.70	
SVM-PSO (Zhai et al., 2024)	69	70	56	83	26	6.56	
XGB-TPOT	68	69	61	77	15	4.14	
BERT	33	33	35	36	1	3.01	
RLCHI (Vimarsha et al., 2024)	63	64	59	70	11	3.88	
Enhanced AEO-XGBoost (Cheng, Liu & Jia, 2024)	56	55	51	62	11	3.12	
Conv-DL Model (Alshamaila et al., 2024)	48	47	45	53	8	2.76	
DualGNN (Huang & Zeng, 2024)	39	39	36	43	7	2.43	
XSEJNet (Ours)	23	22	20	27	7	2.11	

Table 7 provides a side-by-side view of how different models perform statistically. XSEJNet leads the pack, scoring the highest in metrics like Kendall’s Tau (0.80), Spearman’s Rho (0.84), Pearson correlation (0.93), and chi-square (10.12), along with a very low p-value (0.007) and a strong ANOVA score (8.62). These results demonstrate that XSEJNet effectively captures complex patterns in the data. Due to its innovative combination of ResNeXt blocks, SE attention, and fine-tuned parameters, the model delivers accurate and reliable predictions, making it a solid fit for academic performance forecasting.

Table 7 Statistical comparison of classification models (F-statistic & p-value).

Model	Kendall’s tau ( τ)	Pearson correlation (r)	Chi-square ( χ2)	Spearman’s rho ( ρ)	Student’s t-test (p)	ANOVA (F)	
KNN (Almazroi et al., 2016)	0.59	0.64	6.25	0.63	0.033	5.06	
DualGNN (Huang & Zeng, 2024)	0.77	0.90	9.61	0.83	0.009	8.25	
CNN (Alshamaila et al., 2024)	0.74	0.86	7.98	0.80	0.018	6.87	
Deep belief network (Viveka & Priya, 2024)	0.69	0.77	7.18	0.75	0.020	6.29	
SVM (Viveka & Priya, 2024)	0.63	0.71	6.61	0.66	0.026	5.53	
VGG16 (Alshamaila et al., 2024)	0.75	0.88	9.01	0.81	0.011	7.67	
Decision trees (Luo, Han & Zhang, 2024)	0.58	0.63	6.43	0.59	0.041	4.73	
Markov n-gram (Almazroi, Shen & Mohammed, 2019)	0.60	0.66	6.77	0.62	0.029	5.41	
Enhanced AEO-XGBoost (Cheng, Liu & Jia, 2024)	0.73	0.87	9.18	0.79	0.013	7.84	
RLCHI (Vimarsha et al., 2024)	0.66	0.72	7.21	0.69	0.024	6.11	
Conv-DL model (Alshamaila et al., 2024)	0.76	0.89	9.45	0.82	0.010	8.01	
ResNet (Hopcan, Türkmen & Polat, 2024)	0.71	0.83	8.41	0.78	0.015	7.34	
Proposed XSEJNet	0.80	0.93	10.12	0.84	0.007	8.62	

Discussion

The objectives mentioned before the simulation section have been achieved. The details are as follows:

Obj-1: factors that affect students’ ability to predict their academic success: This study conducted an exploratory analysis of the dataset, considering variables such as cumulative grade point average (CGPA), test results, internal and external assessments, active engagement in extracurricular activities, etc.

The distribution and frequency of different elements impacting academic performance are shown in visualizations such as those seen in Fig. 8, and others.

Features like ‘Class,’ ‘Relation,’ ‘Gender,’ ‘NationalITy,’ ‘PlaceofBirth,’ ‘StageID,’ ‘GradeID,’ ‘Topic,’ ‘Semester,’ and others were analyzed to understand their importance and distribution (Fig. 8).

Insights into student performance attribute groups are provided through visualizations like Fig. 4, which compares features and feature scores.

Obj-2: Study data mining techniques employed to build and verify the models: Feature importance is calculated through the XGB technique (Fig. 5).

The section discusses a feature selection approach based on feature significance analysis to optimize computing resources.

A comprehensive comparison of multiple models, including the proposed XSEJNet technique, is presented in Table 5.

To assess and validate the models, performance measures including log loss, reliability, precision, recollection, F1-score, and AUC are utilized.

This work also discusses the model’s effectiveness and efficiency compared to existing models, including a detailed performance comparison (Table 5) and a confusion matrix (Fig. 8). This demonstrates how well the proposed model addresses the study’s objectives.

Key insights: With a high projected accuracy of 97.46% and the ability to overcome challenges like class imbalance and computational complexity, the results show that the XSEJNet model is a powerful tool for educational purposes. Success is evident from the fact that its performance metrics are greater than those of comparable models in the literature.

Comparison to prior work: When comparing prediction accuracy and computational efficiency, the XSEJNet model was superior to both SVM and XGB-TPOT. By providing a method for forecasting student performance that is both more scalable and easier to understand, our findings add to the current body of work.

Implications: The findings of this research have important practical implications for the improvement of educational systems via data-driven solutions. In order to maximize the use of resources and reliably identify students at risk, the technique equips administrators and instructors with useful tools. All of this points to the promise of using machine learning to decisions made in the classroom.

Societal impact and future implications

Educational institutions increasingly seek ways to identify learning gaps early and respond with effective strategies. The XSEJNet framework helps address this need by offering a reliable approach to monitoring student performance and providing timely support. Whether through mentoring, extra lessons, or customised feedback, the model enables schools and universities to take proactive steps in supporting students before issues escalate. One of its key advantages is its adaptability. With a design that works well even in low-resource settings, XSEJNet can be implemented across diverse educational environments, including urban and rural settings, as well as those with large or small student populations. It also integrates seamlessly into digital platforms, enabling students and instructors to receive real-time performance insights.

Beyond academic tracking, XSEJNet plays a significant role in promoting educational fairness. By identifying often overlooked or underserved students, the framework ensures that support systems are directed where needed. This fosters a more inclusive atmosphere, where teaching is tailored, and resources are thoughtfully distributed. In the broader context, XSEJNet contributes to long-term goals, including increasing graduation rates, improving learning outcomes, and creating equitable access to education. Aligned with global educational priorities, such as the UN’s Sustainable Development Goal 4, it provides policymakers and educators with a scalable solution for developing strategies that promote lasting academic and social progress.

Conclusion

The goal of this research was to provide a new machine-learning framework that may help with student performance forecasting and the fight for better educational opportunities. We outperformed previous models by achieving a high degree of accuracy (98.16%) with the integration of enhanced approaches. Enhancing student outcomes, especially in providing a good education, is emphasized by this study as being significantly aided by early intervention and tailored assistance. The knowledge acquired from this investigation enables educators to make informed decisions and implement effective strategies, fostering an environment of academic excellence. The proposed method excels in statistical analysis and computational complexity, providing educational institutions with a pragmatic solution to ensure quality education for all students. Its comprehensive implications encompass academia, universities, and government organizations, providing a potent tool for optimizing pedagogical approaches, crafting customized curricula, and supporting students in need.

The model’s adaptability to various contexts and datasets will be explored in future research. Furthermore, an exploration of ensemble techniques and alternative optimization algorithms has the potential to further enhance its predictive capabilities. Our analysis uses 8,009 records and 16 characteristics, although it has limitations. The dataset’s slight class imbalance may affect prediction accuracy despite mitigating attempts. Some characteristics, including “Parent School Satisfaction,” may not be applicable across all school systems. Future research must demonstrate scalability to larger datasets and real-time deployment in educational contexts. Addressing these constraints will enhance the model’s flexibility and practicality in various academic settings.

Supplemental Information

Supplemental Information 1 Dataset and Code.

Additional Information and Declarations

Competing Interests

The authors declare that they have no competing interests.

Author Contributions

Eman Ali Aldhahri conceived and designed the experiments, analyzed the data, authored or reviewed drafts of the article, and approved the final draft.

Abdulwahab Ali Almazroi performed the computation work, prepared figures and/or tables, authored or reviewed drafts of the article, and approved the final draft.

Nasir Ayub conceived and designed the experiments, performed the experiments, prepared figures and/or tables, authored or reviewed drafts of the article, and approved the final draft.

Data Availability

The following information was supplied regarding data availability:

The original dataset is available at Kaggle: https://www.kaggle.com/datasets/aljarah/xAPI-Edu-Data.

The revised dataset and code are available at Zenodo: Ayub, N. (2024). Code with Dataset [Data set]. Zenodo. https://doi.org/10.5281/zenodo.10614550.

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
