# Peer review of "Regularized multi-path XSENet ensembler for enhanced student performance prediction in higher education"

_PeerJ Computer Science, doi:10.7717/peerj-cs.3032_

## Round 0.1 · original submission · Major Revisions

Dear Authors,

Your paper has been reviewed. Based on the reviewers' reports, major revisions are needed before it is considered for publication in PEERJ Computer Science. The issues you have to fix in your revised version of your paper are mainly the following:

1) You must improve the clarity of the section Introduction by adding the latest relevant literature review to improve understanding and quality.

2) You must compare the performance of your method with state-of-the-art methods relevant to predicting student performance in higher education settings.

3) You should draft the revised version of their manuscript in a more professional and technical style and avoid employing overly colloquial and informal language.

Reviewer 1 ·

Basic reporting

This study proposes the "Regularized Multi-Path XSENet Ensembler" as a novel method for predicting student performance in higher education settings. The research leverages multi-path learning to improve the robustness of performance predictions by combining several sub-models in an ensemble. Regularization techniques are applied to enhance the generalizability of predictions, particularly under varied academic conditions. This approach is aimed at addressing challenges related to high variability in student data and achieving more precise predictions that can assist educational institutions in identifying students at risk and providing targeted support. Further suggestions are:
1. Line number 363 starts with word also, (which don't have any link with other text or content in paper)
2. Authors should avoid words we, our, us, I my and me in research work
3. Figure no 9 and 10 are not clear.
4. Analytical results are missing.
5. Comparative results with other researchers should be included.
6. Impact on society of your research should be mentioned.

Experimental design

Suggestions are:
1. Comparative Evaluation: Include more baseline models, especially ensemble techniques, for a comprehensive evaluation.
2. Interpretability: Provide additional insights into the interpretability of predictions for educational stakeholders.
3. Technical Details: Expand on computational costs, training times, and hyperparameter optimization.
4. Future Implications: Elaborate on real-world applications and potential adaptations across diverse student populations.

Validity of the findings

Authors should validate their results with comparing the results of other researches in the same field.

Additional comments

Other suggestions are:
1. Figures must be neat and clean.
2. Process and outcome shown in figures must be explained in text.

Reviewer 2 ·

Basic reporting

The work is well written but needs improvements on certain aspects:
1. Proof-read the manuscript.
2. Avoid using full forms repeatedly. For example, Deep Learning may be written as DL after using the full form once.
3. Avoid using personal pronouns.
4. Page 13 has unnecessary spaces.
5. Figures 8 and 9 are not clear. Figure 2 needs to be reconstructed.
6. The proposed methodology figure depicts the work well but needs improvement and can be re-made.
7. The discussion section needs more elaboration.
8. Problem statement section should be summarized.
9. Very few works are considered in the literature. Authors can refer: https://link.springer.com/article/10.1007/s42979-024-03118-3

10. Tables 3 and 4 need more discussion.

Experimental design

The work falls within the journal's scope, and the research problem is well-defined. However, the results need to be elaborated, especially the discussion section.

Validity of the findings

The findings are well validated and compared however authors are suggested to add a description of the no of students and the features taken for their dataset and compare them with the the no of students and the features taken by existing work. Also what are the limitations of this work should be mentioned in brief.

Additional comments

No comments.

---

## Round 0.2 · Major Revisions

Your paper has been reviewed. Based on the reviewer's report, major revisions are needed. More precisely, the main issue in your revised version of your paper is the lack of comparison between your method's performance and state-of-the-art methods relevant to predicting student performance in higher education settings. You must must face this point.

Reviewer 1 ·

Basic reporting

following are my observations:
1. Title of the manuscript is appropriate.
2. All the essential information has been given in the abstract.
3. Keywords are missing
4. All figures are cited in text of the manuscript, the figures must be explained properly.
5. Discussion part Should be discussed with results (Images/tables/graphs)
6. Good number of references are incorporated.

Experimental design

The work be impactful for the society.
But, their are many new technologies in the market for such kind of work The authors must explain the reason why they opted this technique. Also, they must compare the result with other existing techniques.

Validity of the findings

the results looks better. But, my suggestion the researchers must apply the similar data on multiple techniques at least 3 to compare these results. Then the comparative work should be explained that may help them to present how their technique is better than others.

Additional comments

Many similar techniques are available in the era. The authors must compare their work with other techniques.

---

## Round 0.3 · Minor Revisions

Dear Authors,
Your paper has been revised. Minor revisions are needed before it is accepted for publication in PEERJ Computer Science. More precisely:

1) You must check the grammar of English and fix all the typos in the paper.
2) The caption of Figure 2 must be expanded to clarify its content better.

**Language Note:** The Academic Editor has identified that the English language must be improved. PeerJ can provide language editing services - please contact us at [email protected] for pricing (be sure to provide your manuscript number and title). Alternatively, you should make your own arrangements to improve the language quality and provide details in your response letter. – PeerJ Staff

Reviewer 1 ·

Basic reporting

The title of paper motivate the new researchers to work in this field.
The authors have incorporated all the suggestions.
I suggest to authors to read the abstract part once more for grammar.

Experimental design

The results of the manuscript are motivating.
Process or working of Figure 2 should be explained

Validity of the findings

Results presented by the authors are motivating

Additional comments

Authors should read the manuscript once more. Many grammatical errors has been seen

---

## Round 0.4 · accepted · Accept

Dear Author,
Your paper has been revised. It has been accepted for publication in PEERJ Computer Science. Thank you for your fine contribution.

Reviewer 1 ·

Basic reporting

The work is impressive

Experimental design

The result are good and impactful. Suggestions have been incorporated by the authors.

Validity of the findings

The comparative results showed that the results are valid.

Additional comments

Paper can be accepted.